# GENSHIN: GENERAL SHIELD FOR NATURAL LANGUAGE PROCESSING WITH LARGE LANGUAGE MODELS

## ABSTRACT

Large language models (LLMs) like ChatGPT, Gemini, or LLaMA have been trending recently, demonstrating considerable advancement and generalizability power in countless domains. However, LLMs create an even bigger black box exacerbating opacity, with interpretability limited to few approaches. The uncertainty and opacity embedded in LLMs' nature restrict their application in high-stakes domains like financial fraud, phishing, etc. Current approaches mainly rely on traditional textual classification with posterior interpretable algorithms, suffering from attackers who may create versatile adversarial samples to break the system's defense, forcing users to make trade-offs between efficiency and robustness. To address this issue, we propose a novel cascading framework called **Genshin** (**Gen**eral **Shi**eld for **N**atural Language Processing with Large Language Models), utilizing LLMs as defensive one-time plug-ins. Unlike most applications of LLMs that try to transform text into something new or structural, Genshin uses LLMs to recover text to its original state. Genshin aims to combine the generalizability of the LLM, the discrimination of the median model, and the interpretability of the simple model. Our experiments on the task of sentimental analysis and spam detection have shown fatal flaws of the current median models and exhilarating results on LLMs' recovery ability, demonstrating that Genshin is both effective and efficient. In our ablation study, we unearth several intriguing observations. Utilizing the LLM defender, a tool derived from the 4th paradigm, we have reproduced BERT's 15% optimal mask rate results in the 3rd paradigm of NLP. Additionally, when employing the LLM as a potential adversarial tool, attackers are capable of executing effective attacks that are nearly semantically lossless. We conduct detailed case analyses using the SHAP interpreter, which could yield insights for systemic enhancements. Lastly, we provide discussions on the architecture of Genshin, underscoring the necessity of each component and outlining the current limitations.

## 1 INTRODUCTION

Masked language modeling (MLM) proposed in BERT (Devlin et al., 2018), which is inspired by the "Cloze" test, reveals the phenomenon of redundancy in natural languages. Although recently the new paradigm of the GPT series developed by OpenAI has made significant progress than the MLM paradigm, the nature of MLM's representative mechanism remains valuable.

For example, "password", "pa*skey" and "$\rho$ass$\omega$Ord" can convey the same concept of "a key to someone's account" to human receivers, but the latter ones are much more difficult for classical pattern-matching machines to identify. This redundancy keeps certain symmetrical invariance in the language space, where a small degree of disturbance on the original text can keep the corresponding meaning of the text intact. The malicious attackers may utilize this to change the sentence's appearance while keeping it human-readable,

Figure 1: Exemplary demonstration of Genshin recovering deliberately altered spam texts and providing classifications and interpretations afterwards.

so that the sentence in disguise may get through security checks of the traditional language model, which mainly relies on rule-based regular expressions at present.

Until recently, LLMs have shown salient potential in symmetry-persistent textual transformation and manipulation, compared with their weakness in logistical inference. This is understandable due to LLMs' natural essence, which generates the most fluent tokens rather than executes solid causal inference. Besides, LMs have shown great discrimination abilities with efficiency on downstream tasks. Finally, IMs provide explanations of the mechanisms inside the black box of the corresponding model. We present an exemplary demonstration of Genshin in Fig. 1.

Due to the intrinsic symmetry of LLMs, there is another potential threat. In the future, attackers may also leverage LLMs not as a shield but as a spear. This kind of attack can be entirely invisible to pattern-matching machines. The race between defenders and attackers leads to infinity, demonstrating the necessity of Genshin.

Our goal is to alleviate the deficiency of current LLMs, by combining the power of median-sized language models (LMs) and interpretable models (IMs). When our framework receives any new textual information, it first calls an LLM to recover it to its original state, if it has been injected with toxic tokens. Then the recovered text is sent to an LM to make accurate predictions. If requested, an IM will be operated to explain the predictions of the LM, providing solid insights and interpretations.

Our main contributions are as follows: (1) A novel framework called Genshin is proposed to provide a general shield for all kinds of adversarial textual attacks. This shield is also capable of becoming a spear in the future; (2) To the best of our knowledge, this is the first attempt to use LLMs as symmetrical recovery one-time plug-ins, aiming to transform the content to its natural status without meaningful information losses; (3) We also leverage LMs and IMs to achieve both efficiency and explainability which current LLMs lack. We visualize our interpreting results to provide insight and enlightening.

## 2 RELATED WORKS

**Language Models**    There are 4 paradigm shifts for language models. In the early days, statistical models based on the Markov assumption (A.A. Markov, 1953) dominated the region of NLP, predicting the next state

with the nearest former state (Zhao et al., 2023). The essence of the Markov assumption is compressible information, where history can be concentrated into constant storage. Second, distributed representations like word2vec (Mikolov et al., 2013a;b) start to mine the hidden features of the textual corpus, introducing the idea of vectorization into NLP territory. Third, pre-trained language models (PTMs) like BERT and GPT-1/2/3 (Radford et al., 2018; 2019; Brown et al., 2020) significantly boost the growth of NLP applications, by separating the training stage and fine-tuning stage. Finally, large language models such as ChatGPT (OpenAI, 2023), LLAMA (Touvron et al., 2023), Claude-3 (Anthropic, 2024), and GPT-4 (Achiam et al., 2023) have emerged with novel abilities that can solve real-world tasks with versatile prompts. LLMs' zero-shot abilities (Kojima et al., 2022) also pave the way for prompt engineering (Liu et al., 2023), optimizing the deficiency of LLMs. For instance, few-shot prompting (Brown et al., 2020) and instructions help LLMs form structural output. Retrieval-augmented Generation (RAG) (Lewis et al., 2020) helps LLMs acquire knowledge and minimize hallucination. Chain-of-thought (CoT) (Wei et al., 2022) and intelligent agents (Park et al., 2023; Team, 2023; Wen et al., 2023) help LLMs cultivate rational minds and stabilize the reasoning process.

**Textual Adversarial Attacks**  Adversarial attacks make LMs vulnerable by creating different kinds of adversarial samples(Wang et al., 2019; Biggio et al., 2013; Szegedy et al., 2013). Textual adversarial attacks can be operated on 3 levels: char-level, word-level, and sentence-level. All textual attacks can be described as some versions of insertion, replacement, or deletion. OpenAttack (Zeng et al., 2020) proposes an open-source textual adversarial attack toolkit, implemented with lots of existing attacking methods. There are also studies for adversarial attacks on LLMs (Shayegani et al., 2023; Weng, 2023). The most well-known adversarial attack is the "magic" prompt, instructing the LLM to forget all safety measures and play a user-specified role to produce harmful content.

**Interpretable Machine Learning**  Interpretable Machine Learning consists of interpretability and explainability (Molnar, 2020). Interpretability means a "white box", where the model's inner mechanism is transparent and understandable to users. There are few classical interpretable models such as the Generalized Additive Model, Decision Tree, Naive Bayes, K-NN, etc (Hastie et al., 2009). But when it comes to neural networks, interpretability withers away when the net scales exponentially according to the scaling law (Kaplan et al., 2020). Explainability aims to provide insight from opaque models, especially neural networks. Model-agnostic methods like LIME (Ribeiro et al., 2016) and SHAP (Lundberg & Lee, 2017) explain classification results by highlighting the importance of different input elements. Instance-based methods explain results by locating the more relevant training sample with crafted influence functions (Han et al., 2020). There are also specialized methods only for interpreting language models (Sun et al., 2021). Attention-based methods aggregate and visualize the attention score toward token importance. Probing explains each layer of neural networks by applying different downstream tasks. Explanation generation explains smaller models with an LLM.

## 3 METHODOLOGY

The textual world is filled with natural or intended noises, where humans might encounter typographical errors or indirectly express their concepts. The LM often fails in these infinite heavy-tailed scenarios causing OOD (Out-of-domain) tokenization problems, leading to an inevitable trade-off between robustness and effectiveness. The LLM has seen them all, on the other hand, it becomes the perfect tool to recover risky information into safe information, ensuring that the LM doesn't require strong robustness to perform well.

Our Genshin framework's workflow shown in Figure 2 includes three stages: (a) **denoising stage (LLM)** where the LLM defender first acts as a one-time recovery tool to denoise the text, from risky information to recovered information; (b) **analyzing stage (LM)** where the LM analyzer can then easily execute

information analysis on the recovered information, including tasks like classification, regression, etc; (c) **interpreting stage (IM)** where the IM interpreter precisely explains the LM's outputs.

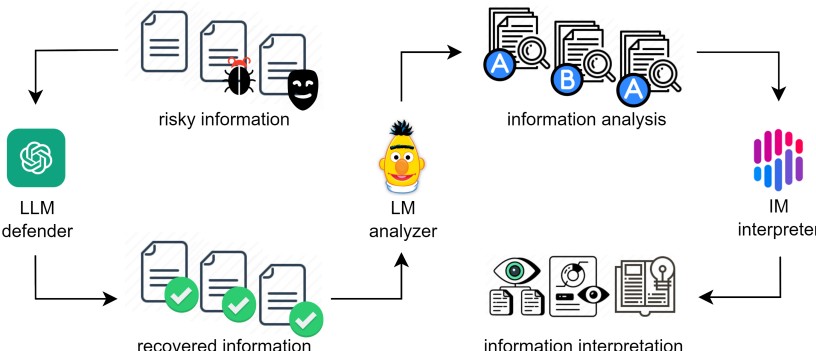

Figure 2: The workflow of the Genshin framework

**Attacking Strategies**  We designed 3 kinds of attacks that might bypass the current system:

- **Char-level disturbance (Char Attacker)**: we randomly disturb each minimal-sized token (characters) in the sentence, with a probability of $\alpha$ (disturbance ratio). This is effective due to redundancy in natural languages.

- **Word-level disturbance (Word Attacker)**: this is similar to char-level disturbance, except happening on words as tokens. For simplicity, the disturbance of characters and words is implemented by random replacement in the following experiments.

- **Similarity-based disturbance (LLM Attacker)**: we prompt LLMs to disturb some words using their synonyms or similar-shaped tokens. In this way, we can provide safety even if the attackers are implementing their attacks through LLMs as well.

**Prompt Design**  We design our attacker and defender prompts with 4 components: task description, demonstration, input, and structural output control. The task description part describes the purpose of the task, like "try to recover the following text to its original state". The demonstration part provides examples of detailed situations, like "there could be adding, deleting, swapping or replacing characters/words". The input part is parameterized as "«text»" ready to be replaced by any new instance in real-time. The structural output control part is crucial, making sure the LLM's output is JSON-parsable, like "output your recovered output text as JSON format".

**Interpreter**  We use SHAP as the interpreter to evaluate token importance in this work. SHAP represents the $i$-th token importance $\varphi_i(f)$ for model $f$ based on Sharpley values, as shown in Eq. 1, where $N$ is the universal set and $S$ stands for all potential combinatorial subsets excluding the $i$-th token. The Sharpley value is calculated as the mathematical expectation of the marginal profit on all combinatorial conditions.

$$\varphi_i(f) = \sum_{S \subseteq N \setminus \{i\}} \frac{|S|!(N - |S| - 1)!}{N!} [\, f(S \cup \{i\}) - f(S) \,]  \tag{1}$$

## 4 EXPERIMENT

In this part, we want to demonstrate the power of LLMs' recovery ability, by attacking the original textual dataset and recovering it later. We measure the targeted LM's accuracy on textual datasets for each state: original state, attacked state, and recovered state.

**Datasets.** We execute our experiments on 2 tasks: sentimental analysis and spam detection. We select 3 datasets from HuggingFace, including *stanfordnlp/sst2* (short as "sst2" in the following content), *dair-ai/emotion* (short as "emotion"), and *Deysi/spam-detection-dataset* (short as "spam-detection").

**Models.** For each task, we select a corresponding pre-trained LM from HuggingFace, including *textattack/bert-base-uncased-SST-2* (short as "bert-base-*") and *mariagrandury/roberta-base-finetuned-sms-spam-detection* (short as "roberta-base-*"). We use GPT-3.5 as the LLM defender's backbone model.

The experimental results are shown in Tab. 1 and Tab. 2. To ensure efficiency, we set a maximum attacking time of 128 for attackers, exceeding that will be considered a failure attack.

Table 1: Results of adversarial attack strategies and recovery experiments on sentiment analysis and spam detection (**disturbance ratio: 0.15**). OAcc: original accuracy (the initial statistical performance for the model on the dataset); AAcc: attacked accuracy (performance after the dataset was attacked); RAcc: recovered accuracy (performance after the dataset was recovered by LLM); RRatio: recovery ratio (RRatio = (RAcc-AAcc)/(OAcc-AAcc)). MAT: median attacking time (how many times the attacker takes to try to make a successful attack).

| Attacker | Model | Dataset | OAcc | AAcc | RAcc | RRatio | MAT |
|---|---|---|---|---|---|---|---|
| char | bert-base-* | sst2 | 0.9766 | 0.1484 | 0.9258 | 0.9387 | 5.0 |
| | | emotion | 0.7734 | 0.1367 | 0.7188 | 0.9141 | 4.0 |
| | roberta-base-* | spam-detection | 0.8789 | 0.3477 | 0.8750 | 0.9926 | 8.5 |
| word | bert-base-* | sst2 | 0.9766 | 0.1836 | 0.5039 | 0.4039 | 30.0 |
| | | emotion | 0.7734 | 0.1680 | 0.5117 | 0.5677 | 10.5 |
| | roberta-base-* | spam-detection | 0.8789 | 0.6211 | 0.7969 | 0.6818 | 128.0 |
| LLM | bert-base-* | sst2 | 0.9766 | 0.1992 | 0.9336 | 0.9447 | 5.0 |
| | | emotion | 0.7734 | 0.2070 | 0.7188 | 0.9034 | 3.0 |
| | roberta-base-* | spam-detection | 0.8789 | **0.0820** | 0.8750 | **0.9951** | 5.0 |
| Average | | | | | | 0.8158 | |

In Tab. 1, the disturbance ratio for char and word attack is set to 0.15 by default. For each original dataset, we sample and construct a subset for experiments consisting of 256 textual instances with text lengths ranging from 50 to 150 characters. We also keep the equilibrium between classes as 128 instances for each class in both classification tasks. The results show that, on average, 81.6% of successful textual attacks can be shielded by the LLM defender. For some special cases like the LLM attacker on the spam-detection dataset, the recovery ratio can be up to 99.5% (nearly perfect).

Different attackers' attacking efficiency varies. MAT (median attacking time) represents the median times the attacker needs to disturb each input text to change the model's output for the whole sampled dataset. MAT for the LLM attacker is the lowest, showing that LLM as an attacker also prevails over classical methods. The word attacker fails frequently and possesses a low RRatio, implying its ineffectiveness in designing for attacking. The high attacking failure rate on the spam-detection dataset implies that the spam features are

far more distributed in the context than sentimental features. The LLM attacker has been revealed with its potential for such difficult tasks.

Table 2: Results of adversarial attack strategies and recovery experiments on sentiment analysis and spam detection (**disturbance ratio: 0.3**).

| Attacker | Model | Dataset | OAcc | AAcc | RAcc | RRatio | MAT |
|---|---|---|---|---|---|---|---|
| char | bert-base-* | sst2 | 0.9766 | 0.1953 | 0.7500 | 0.7100 | 2.0 |
| | | emotion | 0.7734 | 0.1133 | 0.6680 | **0.8402** | 2.0 |
| | roberta-base-* | spam-detection | 0.8789 | 0.4453 | 0.8086 | 0.8378 | 2.0 |
| word | bert-base-* | sst2 | 0.9766 | 0.0156 | 0.4219 | 0.4228 | 6.0 |
| | | emotion | 0.7734 | **0.0078** | 0.3438 | 0.4388 | 3.0 |
| | roberta-base-* | spam-detection | 0.8789 | 0.3633 | 0.5938 | 0.4470 | 28.5 |
| Average | | | | | | 0.6161 | |

In Tab. 2, we set a larger disturbance ratio of 0.3. The results show that the attacking success rate increases (the median attacking time is lower as well), while the recovery ability of the LLM defender decreases because of more information losses. For attackers, this is an inevitable trade-off between attacking efficiency and transmission credibility. It is worth pointing out that the LLM attacker is disturbance-ratio-free due to a lack of controllability from our practical observations, thus we don't reproduce this setting in Tab. 2.

## 5 ABLATION STUDY

### 5.1 DISTURBANCE RATIOS, ATTACKERS, AND DATASETS

To further evaluate recovery ability on different disturbance ratios, attackers, and datasets, we execute our ablation study on three settings: (1) a generated dataset attacked by the char attacker; (2) a standard dataset attacked by the char attacker; (3) a standard dataset attacked by the LLM attacker.

The generated dataset is created by assembling random words into 500 sentences with specific lengths ($\leq$ 100 characters). The standard dataset is sampled from the emotion dataset with 500 sentences with lengths ranging from 50 to 100 characters. We split 500 samples into 5 groups and execute our experiment on each group independently to gather statistical standard deviations.

We utilize the editing distance ratio (EDR), shown in Eq. 2, to calculate the distance between two sentences. EDR is designed to be a normalized Levenshtein distance ranging in $[0, 1]$.

$$\text{EDR}(a, b) = \frac{\text{editing\_distance}(a, b)}{\max(|a|, |b|)} \tag{2}$$

As shown in Eq. 3 and Eq. 4, We further define the average disturbance distance (ADD) as the expectation of all EDRs from each textual pair between the original and attacked dataset, and the average recovery distance (ARD) as the expectation of all EDRs from each textual pair between the original and recovered dataset. $N$ is the dataset size, and $o_i$, $a_i$, $r_i$ represents the $i$-th original, attacked and recovered text accordingly.

$$\text{ADD} = \frac{1}{N} \sum_{i=1}^{N} \text{EDR}(o_i, a_i) \tag{3}$$

$$\text{ARD} = \frac{1}{N} \sum_{i=1}^{N} \text{EDR}(o_i, r_i) \tag{4}$$

The LLM attacker lacks controllability, thus we first make random disturbances with the LLM attacker, calculate their EDR with the original text, and categorize them into the nearest disturbance ratio accordingly. Each ratio category is set to include no more than 500 samples.

The results are shown in Fig. 3. The shadowed areas represent corresponding standard deviations. Note that for the char attacker, ADD is slightly lower than the corresponding disturbance ratio because we have designed the char attacker to skip spaces and punctuations.

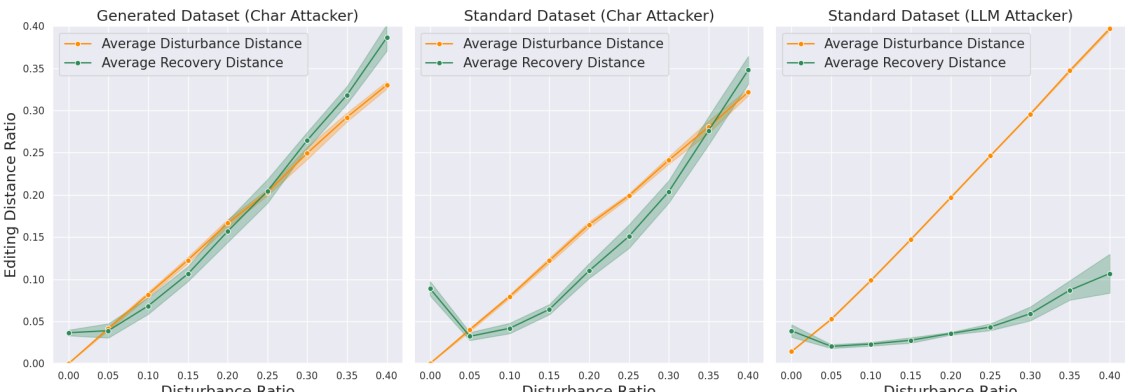

Figure 3: The ablation study results on different disturbance ratios, attackers, and datasets

First, for the generated dataset, the absence of contextual structures in the generated sentences makes the LLM defender nearly impossible to recover anything. Instead, recovering attempts may even introduce excessive noises when the disturbance ratio is $\geq 0.3$. ARD calculated after the LLM defender recovered the attacked dataset, has made no evidential improvements,

Second, for the standard dataset attacked by the char attacker. With a moderate disturbance ratio ranging in $[0.05, 0.35]$, the LLM defender can successfully make restorations for the attacked dataset. Specifically, at the $0.15$ ratio, the LLM defender can recover the highest distance while having the lowest standard deviation. This is consistent with the $15\%$ optimal mask rate conclusion from the mask language modeling (MLM) task in BERT's training.

Furthermore, the ARD curve of the LLM attacker is substantially lower than the char attacker on the standard dataset, which means the LLM attacker has kept most of the semantical information after the disturbance. To conclude, by carefully crafting textual language, we can get a whole new appearance of sentences with minimal information losses, which could overthrow all previous pattern-matching machines. More importantly, this can be done automatically by large language models.

## 5.2 PROMPT ENGINEERING TECHNIQUES

Designing moderate prompts is crucial for downstream task performance. We execute our ablation study on 4 cases: (1) bare prompt: a normal prompt template describes the defensive task; (2) few-shot: the template is provided with a few attacked and recovered examples; (3) JSON parser: the template is instructed to output as parsable JSON format; (4) full prompt: the complete template with few-shot and JSON parser.

The dataset is constructed in the same way as our main experiment, with 512 samples from each original dataset ranging in [50, 150] characters. The char attacker is set with a disturbance ratio of 0.15.

Results are shown in Tab. 3. The few-shot prompting makes improvement more significant when the original RRatio is low, and JSON parser prompting keeps consistency at all conditions. ARD, on the other hand, is improved for all datasets when more prompting techniques are involved, which indicates the LLM defender can recover raw information more accurately with full prompting technical support.

Table 3: The ablation study results on different prompt engineering techniques

| techniques | Dataset | | | | | |
| | sst2 | | emotion | | spam-detection | |
| | RRatio | ARD | RRatio | ARD | RRatio | ARD |
|---|---|---|---|---|---|---|
| bare prompt | 0.7644 | 0.2127 | 0.9244 | 0.1087 | 0.9701 | 0.0903 |
| few-shot | 0.8702 | 0.1298 | 0.9593 | 0.1024 | 0.9627 | 0.0847 |
| JSON parser | **0.8846** | 0.0882 | **0.9651** | 0.0686 | **1.0075** | 0.0560 |
| full prompt | 0.8606 | **0.0836** | 0.9477 | **0.0644** | 0.9851 | **0.0552** |

## 6 CASE STUDY

In this part, we implement interpretable algorithms and demonstrate how Genshin can help provide a better understanding of the attacking mechanism in detailed cases.

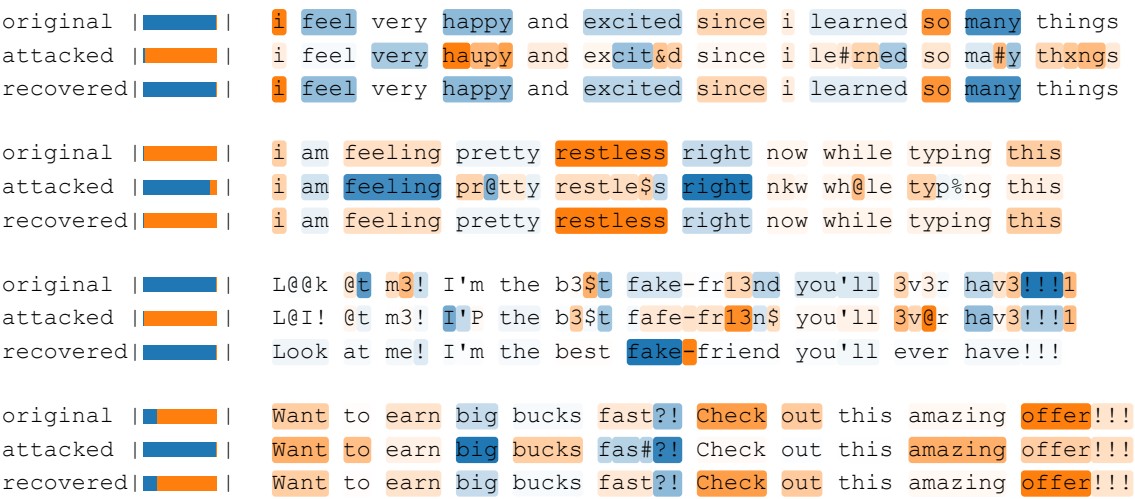

Figure 4: Our interpreting results with the char attacker.

Our exemplary visual interpreting results on 4 cases are shown in Fig. 4. The first 2 cases are from the emotion dataset, and the last 2 cases are from the spam-detection dataset. For each case, we demonstrate 3 rows with the original, attacked, and recovered text. All attacks are executed by the char attacker. For each row, the rectangle bar on the left visualizes the model's probability output. The blue color represents the label "POSITIVE" or "HAM", and the orange color represents the label "NEGATIVE" or "SPAM". The opacity of the background color is linearly related to the token importance value derived from SHAP.

For example, in the 2nd case, the word "pretty" and "restless" is attacked, becoming "pr@tty" and "restle$s". The model is successfully fooled by the attacker making its judgment by "feeling ... right". In the 3rd case, the original text is almost completely unrecognizable to humans, and the LLM defender recovered it perfectly. These interpreting results can provide insight into bad case refinement and further improvement guidance for the task.

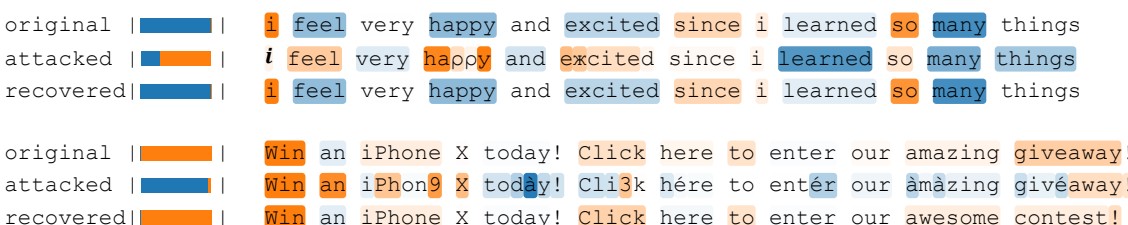

Figure 5: Our interpreting results with the LLM attacker.

In our main experiments, the LLM attacker works much better on the spam-detection dataset compared with other attackers. We demonstrate 2 cases in Fig. 5 attacked by the LLM attacker. All other settings remain consistent. As presented, the attack results by the LLM attacker are more human-readable because of the elaborate usage of similar-shaped tokens.

## 7    DISCUSSION

**Enhancing Robustness through Large Language Models**    The augmentation of our framework, Genshin, with Large Language Models (LLMs) is driven by the requirements for improved robustness, which traditional models like BERT fail to offer due to their reliance on tokenization. Textual attacks can easily disrupt these tokenizers, leading to ineffective content moderation. LLMs enhance robustness comprehensively, mitigating the impacts of such adversarial attacks. Additionally, incorporating LLMs as a one-time plugin allows the median-sized Language Models (LMs) in our system to focus on maximizing accuracy and performance in content analysis and classification tasks. Let's ponder this: "What if the attackers also leverage an LLM to create adversarial samples?" In such cases, a system without an LLM shield layer will be defenseless.

**The Necessity of Interpretable Models**    In Genshin, the integration of Interpretable Models (IMs) addresses the opaqueness of traditional machine learning outputs. Traditional models often struggle to provide insights into the exact location and nature of issues in detected sensitive or problematic text. Our IMs not only identify such text but also highlight specific segments for further review, significantly aiding content moderators. This increases the transparency and trustworthiness of the model's outputs, leading to more efficient and precise content moderation.

**The Role of Median-sized Language Models as an Intermediate Layer**    In the architecture of Genshin, LMs serve as an intermediate layer, rather than directly applying LLMs for content review followed by IMs. This design is primarily driven by computational efficiency and cost-effectiveness. IMs such as LIME and SHAP require thousands of repetitive calls from previous language models. LMs, being tremendously less resource-intensive than LLMs, can be invoked multiple times with minimal cost. As a result, LMs act as an efficient intermediate layer, handling the majority of text classification tasks by pinpointing key information. In contrast, the high computational cost of LLMs necessitates their limited use. So in our framework, LLMs will only be called once per content review cycle.

**Limitations of Genshin**  While our Genshin framework forms a general shield against all attackers in practice, there are clear limitations beyond its reach. One limitation is the lack of controllability of the LLM attacker, such as creating a large disturbance or a specified disturbance ratio attack. Second, the general LLM defender prompt could be inexperienced for domain-specific tasks like medical records, so including Retrieval-augmented Generation (RAG) into the LLM defender could further enhance the model's robustness. Third, Genshin can only recover textual information while the world is full of multi-modal information, which can restrict its applications for more general purposes.

## 8 CONCLUSION AND FUTURE WORK

In this work, we propose a novel framework called "Genshin", to make trade-offs between LLMs, LMs, and IMs. Our experiments demonstrate the current flaws of classical models and the effectiveness of the Genshin framework. In our ablation study, we successfully reproduced the optimal 15% mask rate results of BERT, previously established in the 3rd paradigm of NLP. Additionally, our experiments with the LLM as a potential attacking tool reveal that attackers can execute successful strategies nearly semantically information lossless. This dual application highlights the robustness and versatility of the Genshin framework in addressing both defensive and offensive scenarios in language processing. In the case study, we interpret multiple cases providing insight for system improvement. Finally, we have discussed the design of Genshin, the necessity of each component, and the limitations at present.

For future work, we want to explore the controllability of the LLM attacker and more prompting skills. Next, novel interpretable algorithms might provide data insights into how the attacks take effect and a deeper understanding of how the model operates. Genshin is also promising to be applied to Optical Character Recognition (OCR) or Automatic Speech Recognition (ASR) results, recovering mistakes made by preceding models. Although we mainly focus on the defense of median-sized models, the safety of LLMs is still worth exploring. With the growth of multi-modal LLMs, how to recover an edited photo, recording or video remains a critical area of research.

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
