# OpenReview forum: "Genshin: General Shield for Natural Language Processing with Large Language Models"
_ICLR.cc/2025/Conference — ICLR 2025 Conference Withdrawn Submission_

### Official Review · Reviewer_CGUr · 2024-10-27

**Soundness:** 3
**Presentation:** 3
**Contribution:** 2
**Rating:** 3
**Confidence:** 4

**Summary:**

This paper proposes a defense method using LLMs as a barrier to resist adversarial text attacks, leveraging large language models to recover adversarial samples and make AI systems more secure. Their approach combines the capabilities of medium-sized language models (LMs) and interpretable models (IMs). If needed, the IM can be utilized to explain the predictions of the LM.

**Strengths:**

By utilizing large language models (LLMs) as symmetrical recovery tools, the Genshin framework effectively defends various adversarial text attacks while maintaining semantic consistency.

The Genshin framework combines the efficient prediction capability of medium-sized language models (LMs) with the interpretability of interpretable models (IMs), addressing the current limitations of LLMs in transparency. Using interpretable models to explain black-box defense systems is very innovative.

**Weaknesses:**

Weakness 1: As an LLM defensive work, there is no discussion of mainstream LLM robustness work after 2023, such as PromptBench[1], etc.

Weakness 2: The core method is too simple. It just rewrites the adversarial sample using the LLM. It is essentially a prompt adjustment work and has limited innovation.

Weakness 3: Moreover, I think interpretable models play a meaningless role in black-box defense, and there is no experiment to discuss whether they can improve defense performance.

Weakness 4: Lack of experiment study, they did not discuss the new attack method but only ran the bert-attack(It is a very old method)


[1] Zhu K, Zhao Q, Chen H, et al. Promptbench: A unified library for evaluation of large language models[J]. Journal of Machine Learning Research, 2024, 25(254): 1-22.

**Questions:**

Q1 The interpretability methods do not seem to provide any additional benefit for defense. Is there any experiment proving that interpretability methods can better defend against adversarial attacks?

Q2 There is no comparative experiment; why not compare with existing adversarial defense methods to highlight your own advantages?

Q3 If the defense method is completely white-box, are there any attack methods that can bypass the defense? This also needs to be discussed.

---

### Official Review · Reviewer_mgqc · 2024-10-30

**Soundness:** 1
**Presentation:** 2
**Contribution:** 1
**Rating:** 3
**Confidence:** 4

**Summary:**

This paper introduces Genshin, a framework designed to defend NLP systems against adversarial attacks by leveraging LLMs as one-time recovery tools to revert manipulated texts to their original state. The reverted text is then classified using mid-sized language models, and SHAP is employed for interpretability in case studies. Experiments conducted on three datasets indicate that the proposed approach effectively reverts manipulated texts.

**Strengths:**

The proposed method is straightforward, and the presentation is clear.

**Weaknesses:**

There are important experimental details missing that make the study challenging to reproduce. For instance, the defense prompt and detailed attacker settings are not provided.

Some aspects of the experimental setup also raise questions:
- It’s unclear why state-of-the-art attack methods were not employed, as the authors instead used three attack strategies involving “random replacement.”
- Additionally, it appears the attackers were tested on the vanilla LM to find adversarial examples and the LLM is then used to revert such changes. In a real-world scenario, attackers would engage directly with a system that has built-in defenses. As such, the current setup leaves the actual effectiveness of the defense and its application in real-world systems somewhat unclear.

The motivation for using GPT-3.5 as a defense tool, while relying on LMs for the main tasks, is also not fully addressed, particularly considering the high cost of LLM inference. Additionally, the study does not compare Genshin with alternative lightweight defenses, such as those proposed by [1] and [2], which would serve as baselines.

- [1] Wang et. al. 2021, Natural language adversarial defense through synonym encoding
- [2] Jia et. al. 2019, Certified Robustness to Adversarial Word Substitutions

**Questions:**

- What are the AAcc and RAcc using state-of-the-art attackers?
- What are the RAcc if the attacker is aware of the defense and directly engage with it?
- How is the proposed method compare with baselines?

---

### Official Review · Reviewer_c1su · 2024-11-03

**Soundness:** 2
**Presentation:** 2
**Contribution:** 2
**Rating:** 5
**Confidence:** 3

**Summary:**

The paper proposes Genshin, a cascading framework designed to defend against adversarial textual attacks on language models. It integrates three components: LLMs for one-time text recovery, median language models for analysis, and interpretable models for transparency. Genshin is validated on sentiment analysis and spam detection, demonstrating substantial resilience against various levels of textual disturbance.

**Strengths:**

- Originality

The paper presents a unique use of LLMs as recovery agents, using them as one-time plug-ins in an adversarial context. This approach differs from standard uses of LLMs for transformation or classification.

- Significance

By demonstrating a framework that can handle adversarial attacks, Genshin has potential applications in high-stakes domains like spam detection, security, and sentiment analysis, making the work relevant and impactful.

**Weaknesses:**

- The reliance on LLMs as recovery tools introduces significant computational cost, which may limit the scalability of Genshin for real-time or resource-constrained applications. LLMs (e.g. Llama, Vicuna, GPT) themselves should be robust enough to the token-level perturbations. I do not see the necessity to use LLMs as an intermediate agent for input recovery and send the input to an LM (i.e. BERT or RoBERTa).

- The experimental setting is limited in terms of tasks and models. This paper did not justify the necessity of using an LM for inference while LLMs are available, which makes the experiment setting a bit confusing where only LMs are being evaluated.

**Questions:**

- If LLM is available in the framework, why is it not directly used for downstream task? Are LLMs robust against the token-level perturbations used in this paper?

- Are there strategies to mitigate the computational demands of the LLM recovery stage, perhaps through a selection mechanism that applies LLMs selectively?

- How feasible is the implementation of Genshin in real-time environments? Would latency impact its performance in high-frequency applications like fraud detection?

---

### Official Review · Reviewer_ZZ3G · 2024-11-04

**Soundness:** 2
**Presentation:** 1
**Contribution:** 2
**Rating:** 3
**Confidence:** 4

**Summary:**

In this work, the author advocate for a defensive framework, Genshin, which leverages LLMs as one-time recovery tools to restore the attacked text against adversarial textual attacks. Genshin manages information in three tiers of processing: denoising with an LLM to get rid of adversarial noise, analysis with a mid-sized LM, and interpretation by an IM to explain the outputs. In applications like sentiment analysis and spam detection, Genshin restores over 80% of the data that are under heavy disturbance.

**Strengths:**

The figure appears clear and well-presented.

**Weaknesses:**

1. The paper’s writing needs substantial improvement. For example, when "IM" first appears, it lacks an explanation (though I understand it stands for "interpretable model"), and similar issues appear throughout the paper. Additionally, the authors need a clear statement of their objectives and a concise summary of the key contributions in both the abstract and introduction. Without this, it is challenging to understand the authors' intentions, as I only understood their goals upon reaching the experiment settings. I strongly suggest the authors revise this version for clarity.

2. The novelty of the paper is limited. Essentially, the authors use an LLM to restore perturbed adversarial text to defend against potential attacks for a BERT-like model, relying on a strong assumption that the attack perturbations are at the character level, which the LLM can detect and recover. However, many other adversarial attacks do not rely on character-level perturbations. Additionally, this approach lacks any performance guarantee, whereas a certified defense approach is generally more favorable in current research.

**Questions:**

Have you consider other adversarial attacks beyond char level?

---

### Official Review · Reviewer_7iq6 · 2024-11-06

**Soundness:** 1
**Presentation:** 1
**Contribution:** 1
**Rating:** 1
**Confidence:** 5

**Summary:**

This paper proposes a General Shield for Natural Language Processing with Large Language Models (Genshin), utilizing LLMs as defensive one-time plug-ins. It includes three stages: 1) denoising stage as a recovery tool to denoise the text, from risky information to recovered information, 2) analyzing stage (LM) where the LM analyzer can then easily execute information analysis on the recovered information and 3) interpreting stage (IM) where the IM interpreter precisely explains the LM’s outputs. It tested 3 attacks such as Char-level disturbance, • Word-level disturbance and Similarity-based disturbance. On the sentimental analysis and spam detection tasks, it shows its great potential over previous bert-base or roberta-based methods.

**Strengths:**

originality: I did not find any originality. I did not find any meaningful contribution in this paper. The used tools are all existing and the proposed method can not prove its usefulness with limited evaluation on simple spam detection datasets.

quality: The comparison is unfair (bert as baselines VS ChatGPT3.5).

clarity: this paper is easy to understand.

significance:  I did not find any significance of this work because the whole pipeline does not make sense to me at all.

**Weaknesses:**

There are so many weaknesses in this paper that I think the authors do not understand what makes a 'standard' paper, such as evaluation, comparison, and motivation. I suggest the authors to read more papers published at ICLR to learn how to write a paper.

The name is "General Shield for Natural Language Processing with Large Language Models" but I did not find any evidence to support the claim of "general".

The contribution is not enough to claim as General Shield for Natural Language Processing when you only test on two simple classification problems.

**Questions:**

1. What does this claim mean? "Utilizing the LLM defender, a tool derived from the 4th paradigm, we have reproduced BERT’s 15% optimal mask rate results in the 3rd paradigm of NLP." Those are totally different targets, and the 15% optimal rate is meaningless.

---

### Note · Authors · 2024-11-28

I have read and agree with the venue's withdrawal policy on behalf of myself and my co-authors.